# SIMPLICITY BIAS LEADS TO AMPLIFIED PERFORMANCE DISPARITIES

## ABSTRACT

The simple idea that not all things are equally difficult has surprising implications when applied in a fairness context. In this work we explore how "difficulty" is model-specific, such that different models find different parts of a dataset challenging. When difficulty correlates with group information, we term this *difficulty disparity*. Drawing a connection with recent work exploring the inductive bias towards simplicity of SGD-trained models, we show that when such a disparity exists, it is further *amplified* by commonly-used models. We quantify this *amplification factor* across a range of settings aiming towards a fuller understanding of the role of model bias. We also present a challenge to the simplifying assumption that "fixing" a dataset is sufficient to ensure unbiased performance.

## 1 INTRODUCTION

Without actually training, understanding *what* a model will find challenging is far from trivial. A certain dataset may be hard for one model but not for another (Wolpert & Macready, 1997). For a given model, two classes may be easily separable, while for another they may be hard to distinguish. Given this, it follows naturally that "difficulty" is a function of both *data and model*, such that we can't properly account for difficulty by analyzing the dataset alone.

In the context of fairness in machine learning, for a given task, a data-model pair may be more difficult for one social group than another, leading to disparate impact (Barocas & Selbst, 2016). For example, Buolamwini & Gebru's 2018 audit of commercial image recognition systems finds

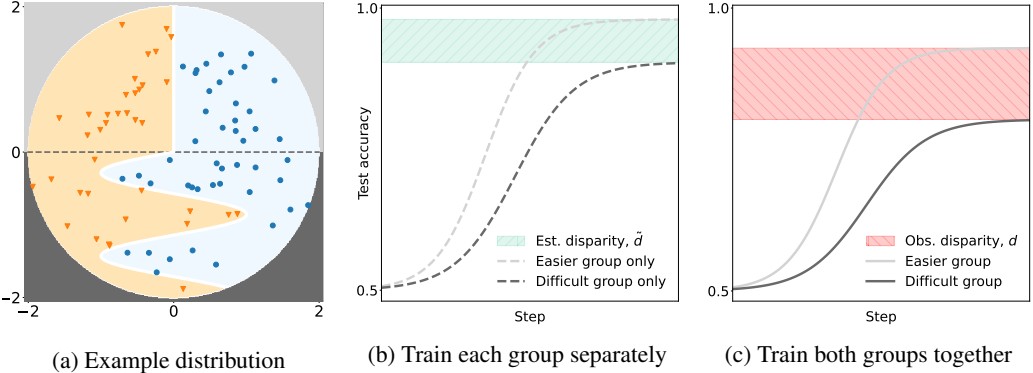

(a) Example distribution  (b) Train each group separately  (c) Train both groups together

Figure 1: **What is difficulty amplification?** **(a)** Consider a binary classification of circles and triangles. Above $y = 0$ (light gray background) we have a simple group which is linearly separable. Below $y = 0$ (dark gray) we have a more complex group with a non-linear decision boundary. **(b)** Illustration of test accuracy when training on the simple group only (light gray) and the complex group only (dark gray). As expected we obtain better accuracy on the simple group. **(c)** However, when training on both groups at once, our model exacerbates the difference: the observed accuracy disparity $d$ (height of pink area) exceeds the estimated accuracy disparity from individual group training $\tilde{d}$ (height of green area). When $d > \tilde{d}$, we call this **difficulty amplification**.

they exhibit worse accuracy for darker-skinned women than for any other group. Typically, accuracy disparity of this kind is attributed to either under-representation of certain groups or spurious correlations between group information and the variable of interest. In this work, we further show that—even with perfectly balanced data and in the absence of correlations between group labels and class labels—trained models can find certain groups harder than others. Crucially, group difficulty is not always predictable from a dataset audit, providing key evidence for the necessity of a complementary post-training model audit.

Having identified model-specific disparities in the post-dataset-audit setting, we turn to the role of the model itself. We show that implicit bias of certain model classes towards simple functions (Arpit et al., 2017; Kalimeris et al., 2019; Rahaman et al., 2019; Valle-Perez et al., 2019; Shah et al., 2020) further amplifies disparity: When a model finds one group easier than another, its bias towards the easy group leads to greater than expected performance disparity after training (see fig. 1). We show that *difficulty amplification* is highly sensitive to model architecture, training time, and parameter count. Seemingly innocuous design decisions, such as whether to use early stopping, can have a significant impact on the amount of amplification and consequently the performance disparity.

**Contributions**

1. We identify **difficulty disparity**, a pervasive phenomenon that persists in the post-dataset-audit setting: using data with perfect representation and without spurious correlations.

2. We introduce **difficulty amplification factor** to quantify how much a model exacerbates difficulty disparity.

3. We **empirically evaluate** how choices including model architecture, training time, and parameter count impact difficulty amplification.

**Paper structure.** In § 2 we provide background and related work. We begin in § 3 by evaluating variability of dataset difficulty across models. In § 4 we formalize difficulty disparity and amplification and design a synthetic task to isolate it. In § 5 we show how factors such as model architecture, scale, training time and regularization impact amplification. After a real-world example using Dollar Street in § 6, we discuss the fairness implication of our work in § 7 before concluding in § 8.

## 2 BACKGROUND AND RELATED WORK

### 2.1 BIASED DATASETS AND BIASED MODELS

*Bias* in ML systems arises from many sources. At the most basic level, a dataset itself is biased if certain groups are under-represented (Stock & Cisse, 2018; Hendricks et al., 2018; Yang et al., 2020; Menon et al., 2021). Proposals to rectify under-representation include actively collecting more data for marginalized groups (Dutta et al., 2020), under/oversampling or reweighting (Byrd & Lipton, 2019; Sagawa et al., 2020; Idrissi et al., 2022; Arjovsky et al., 2022) during training, and optimizing for worst-group (as opposed to average) accuracy (Sagawa et al., 2019). Other recent work has suggested fine-tuning on an explicitly balanced set (Kirichenko et al., 2022).

Alternatively, datasets can reinforce harmful associations (Goyal et al., 2022b), both due to sampling error and by inadvertently capturing an undesirable association that is present in society. Bucketed as "spurious correlations" (Muthukumar et al., 2018; Wang et al., 2019; Sagawa et al., 2020) or "shortcuts" (Geirhos et al., 2020), a large body of fairness work seeks to train models that learn some true function invariant to the spuriously correlated feature. Both under-representation and spurious correlations dominate the fairness literature landscape, though in both cases the onus is squarely on the data.

### 2.2 BEYOND SPURIOUS CORRELATIONS

There is increasing focus on the model itself, independent of the role of data (Hooker, 2021), of which *bias amplification* is a perfect example (Zhao et al., 2017; Wang & Russakovsky, 2021). Here, a small correlation in the training set is amplified into a larger correlation at test time. Models don't just replicate the bias in the data, but exacerbate it. In empirical experiments evaluating bias amplification, Hall et al. (2022) suggest that if group membership is easier to identify than class

membership, models prefer to use the spurious correlation. Hall et al. report no bias amplification in the case where there is no spurious correlation present in the data: this is expected given the definition of amplification involves a multiplication of an existing data bias.

Other subtle biases have been identified in the balanced data setting. Leino et al. (2019) show that models trained with SGD overly rely upon moderately spuriously-correlated features if they are sufficiently numerous relative to the size of training set. Khani & Liang (2020) find that adding feature noise equally across groups induces disparity, a fact that can also be attributed to the relative difficulty of group information versus the desired target. Khani & Liang (2021) find removing spurious features can disproportionately lower performance on certain groups, and argue (as do we) that a balanced dataset is not a sufficient guard against biased performance. Mannelli et al. (2022) use teacher-student networks to show subtle properties such as differences in group distance from the overall mean and differences in group variance are sufficient to induce biased outcomes in the absence of spurious correlations.

### 2.3 Inductive bias towards simplicity

Numerous recent works have identified the tendency for SGD-trained models to prioritize simple data points during training, resulting in simple functions being learned before more complex ones (Arpit et al., 2017; Kalimeris et al., 2019; Valle-Perez et al., 2019). Jo & Bengio (2017) show that convolutional networks are overly dependent on surface-level statistical properties of images, such that applying a Fourier filter to the training set is sufficient to radically degrade test performance. Rahaman et al. (2019) also show a bias towards low-frequency functions, learning these simpler functions before more complex, higher-frequency examples. Though often framed as a positive, allowing neural networks to learn functions that generalize well by applying Occam's razor, Shah et al. (2020) cites this simplicity bias as potentially harmful, at root causing both vulnerability to adversarial attacks and over-reliance on spurious correlations. Dagaev et al. (2021) reserve as much skepticism for overly simple solutions as for overly complicated, arguing that excessively simple solutions are likely to rely on potentially harmful "shortcuts" (Geirhos et al., 2020). Sagawa et al. (2020) find that increasing model size yet further pushes the model to rely upon spuriously-correlated features where they carry more signal than the intended features. In our work, we show how this inductive bias toward simplicity manifests as disparate outcomes when there is a perceived complexity difference between groups.

## 3 Prelude: Data difficulty is model-specific

This work is predicated on a simple, perhaps even obvious idea: that certain facets of a dataset may be more or less difficult to a given model. Whether individual data points, classes, or social groups, identifying what will be difficult to a model isn't always apparent ahead of time, nor need it align with human intuition. As an illustration, we begin by investigating difficulty difference *between classes*, though in following sections we will transition to a focus on social groups.

**Measuring class difficulty.** We measure *difficulty* using cross-validated test accuracy over the full dataset, i.e. after merging the source train and test sets (see appendix B). We train a classifier on coarse-grained CIFAR-100 (Krizhevsky et al., 2009), after which we evaluate binary classification accuracy for each pair of labels by masking irrelevant outputs before the softmax layer. Figure 2a shows how ResNet-18 (He et al., 2016) yields variable test accuracy between classes. For example, this model achieves near perfect test accuracy on *flowers/aquatic mammals*, and substantially lower performance on *non-insect invertebrates/insects*. Crucially, this view of difficulty is just that of this specific model. Were we to perform this evaluation on a different model, our results would differ.

**Model specificity.** To confirm this, we repeat the experiment using the following models: an SVM with an RBF kernel; a 3-layer and a 5-layer fully-connected network; LeNet, a simple CNN (Lecun et al., 1998); AlexNet, a more complex CNN (Krizhevsky et al., 2012) and a fully-connected layer over pre-trained representations extracted from ResNet-50 trained using SimCLR (Chen et al., 2020) on ImageNet-1K (Russakovsky et al., 2015), and those of RegNet-128Gf (Radosavovic et al., 2020) trained with SwAV (Caron et al., 2020) on 1 billion public images from Instagram (Goyal et al., 2022a) (see appendix C for a full description).

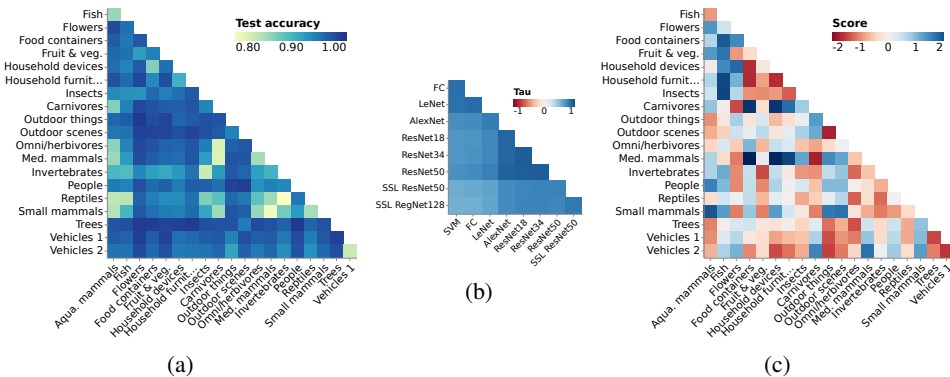

Figure 2: **(a)** ResNet-18 finds certain classes of CIFAR-100 more difficult than others, according to binary test accuracy. **(b)** Rank ordering of difficulty varies across models, measured here by Kendall's $\tau$. **(c)** 1D PLS projection of binary accuracies onto cosine distance between class mean vectors. Red cells—pairs where increasing inter-class distance decreases accuracy—demonstrate **there is no clear way to identify what a model will find difficult without training.**

Figure 2b shows how the rank order correlation of pairwise difficulties varies between models, as measured by Kendall's $\tau$. Specifically, between the 3-layer FC model and LeNet, there is high (but not perfect) $\tau$, indicating that broadly what LeNet finds difficult so too does the FC network. In contrast, between the RBF SVM and a linear layer on RegNet-128Gf representations, there is much lower (though not zero) $\tau$, indicating that the pairwise ordering varies considerably. This aligns with recent work (Hacohen et al., 2020) showing that the difficulty of individual data points is shared across random initializations of the *same model architecture*, but difficulty is only partially consistent *across architectures*.

**Data difficulty is not model difficulty.** We test this using the partial least squares (PLS) analysis shown in fig. 2c. PLS attempts to find low-dimensional projections of both the input and output variables such that their covariance is maximized. We apply PLS to determine how much a *data-only* difficulty measure can explain a *model+data* measure, where the data-only measure is the rank cosine distance between the input data class means, and the model+data measure is the rank test accuracy (see appendix A). If model difficulty was purely a function of data difficulty, we would expect PLS to find a well fitting linear regression model. Instead, PLS finds a near-zero fit ($R^2 = 0.058$). From the 1D projection, we see that for some classes (in blue, e.g. *carnivores/food containers*), increasing inter-class distance tends to increases binary accuracy, though for many class pairs (in red, *outdoor scenes/outdoor things*) the opposite is true.

**Summary.** *There is no clear difficulty ordering that is consistent between all model classes. What a model finds difficult is not solely a function of the data.*

## 4    NEURAL NETWORKS PRIORITIZE "EASY"

Having established that difficulty is a function of both model and data, we now measure how difficulty affects performance disparity and quantify the effect of the simplicity bias.

### 4.1    DEFINITIONS

Let $\mathrm{acc}(X, \mathbf{y}, \mathcal{M})$ be the cross-validated test accuracy on the classification dataset $(X, \mathbf{y})$ of a model $\mathcal{M}$, and $\mathcal{M}_{X,\mathbf{y}}$ be a model trained (with cross-validation) on $(X, \mathbf{y})$. Given two groups $\alpha$ and $\beta$, let $(X_\alpha, \mathbf{y}_\alpha)$ and $(X_\beta, \mathbf{y}_\beta)$ denote corresponding slices of the dataset, such that a model trained only on group $\alpha$ is $\mathcal{M}_{X_\alpha,\mathbf{y}_\alpha}$.

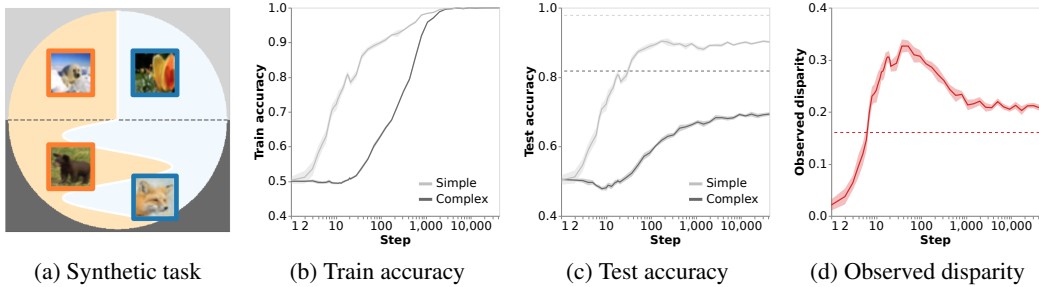

|(a) Synthetic task | (b) Train accuracy | (c) Test accuracy | (d) Observed disparity |

Figure 3: Accuracy and disparity for ResNet-18. **(a)** Binary classification (orange / blue) comprising two groups. Easy group (light gray) is high-accuracy pair *flowers/aquatic mammal*. Difficult group (dark gray) is low-accuracy pair *medium mammals/carnivores*. **(b)** Train accuracy for complex group is learned slower but both groups reach perfect train accuracy. **(c)** However, test accuracy for complex group is persistently lower. Dashed lines are binary accuracy from single-group training. **(d)** Observed disparity $d$ peaks early in training but large gap persists after training. Red dashed line is estimated disparity $\tilde{d}$: observed disparity above this line indicates *amplification*. Shaded area is standard error over 10 runs each with different train/test split.

**Estimated difficulty disparity**. First, we define the estimated difficulty disparity $\tilde{d}$ as the difference in accuracy between a model trained and evaluated on each group in isolation,

$$\tilde{d} = \left| \mathrm{acc}(X_\alpha, \mathbf{y}_\alpha, \mathcal{M}_{X_\alpha, \mathbf{y}_\alpha}) - \mathrm{acc}(X_\beta, \mathbf{y}_\beta, \mathcal{M}_{X_\beta, \mathbf{y}_\beta}) \right|. \tag{1}$$

**Observed difficulty disparity**. Second, the observed difficulty disparity is the difference in accuracy between groups on a model trained on both groups,

$$d = \left| \mathrm{acc}(X_\alpha, \mathbf{y}_\alpha, \mathcal{M}_{X, \mathbf{y}}) - \mathrm{acc}(X_\beta, \mathbf{y}_\beta, \mathcal{M}_{X, \mathbf{y}}) \right|. \tag{2}$$

**Difficulty amplification**. Finally, if the model trained on both groups exhibits worse disparity than when trained in isolation, $d > \tilde{d}$, we say that the model exhibits difficulty amplification. Over many model runs, groups, or samples from the dataset we can define an amplification factor $k = d/\tilde{d}$.

In practice, calculating amplification is a two-stage process. First, we train $N$ randomly-initialized models on each group in isolation, and compute the average cross-validated test accuracy. Between each pair of groups, we calculate the estimated difficulty disparity $\tilde{d}$. Second, we train a new set of $N$ models on the full dataset including all groups, and compute average test accuracy broken out by group. For each group pair, we calculate the observed difficulty disparity $d$.

## 4.2 SIMULATING DIFFICULTY DISPARITY WITH CIFAR-100

To measure difficulty disparity in a controlled setting, we design a task based on CIFAR-100 that is group-balanced and absent correlations between group labels and target labels. We extract the binary test accuracies for each pair of coarse classes and treat their pairwise differences as estimated difficulty disparity $\tilde{d}$. We let group $\alpha$ be the class pair $y_0^\alpha, y_1^\alpha$ with the highest accuracy and $\beta$ be the lowest $y_0^\beta, y_1^\beta$. To simulate a binary classification task with two differently-difficult groups, we stitch these pairs together into a single binary task, where $y_0 = \{y_0^\alpha, y_0^\beta\}, y_1 = \{y_1^\alpha, y_1^\beta\}$. See fig. 3a. Finally, we train $N$ models on this task and calculate observed difficulty disparity $d$.

## 4.3 RESULTS

In fig. 3b we see the training accuracy of the simple group $\alpha$ improves much more rapidly than the complex $\beta$, though both groups reach perfect train accuracy eventually. In contrast, the test accuracies in fig. 3c remain notably different at convergence, with the model displaying lower accuracy on the complex group. This gap is the observed accuracy disparity shown in Figure 3d. Here, we observe that observed disparity peaks after just a few steps, before a slight decline to a plateau.

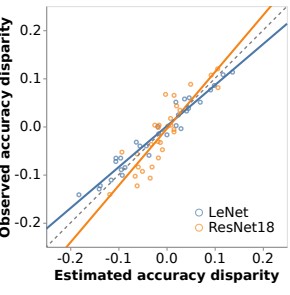
(a) Observed vs. estimated

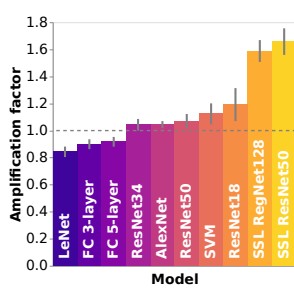
(b) Amplification factor

Figure 4: Observed difficulty disparity as a function of estimated difficulty disparity. **(a) ResNet-18 amplifies difficulty disparity but LeNet attenuates.** Points are sampled tasks with varying estimated difficulty disparities. Solid line is linear regression. Dashed grey line is amplification factor $k = 1$. **(b) Model architecture impacts difficulty amplification factor.** See fig. S4 for $R^2$ and fig. S3 for raw test accuracies.

Evidencing our difficulty amplification claim, observed disparity remains higher than estimated disparity we would have expected from separate training (the red dashed line). Replications of this experiment on both Fashion MNIST (Xiao et al., 2017) and EMNIST Letters (Cohen et al., 2017) show similar results (see fig. S1 and fig. S2).

**Summary.** *This simple experiment shows that models trained across groups with different difficulty do prioritize the simpler group, leading to an outsized observed disparity, primarily driven by under-performance on the more difficult group.*

## 5 AMPLIFICATION FACTOR VARIES ACROSS MODELS

To compute an amplification factor, we repeat the above experiment using different pairs of classes with different estimated disparity, and compute their observed disparity after combined training. We retrain on 30 sampled pairs of label pairs, recomputing both estimated and observed difficulty disparity, and apply OLS linear regression to estimate the *amplification factor* (see appendix E for details). This method can easily be applied to any dataset annotated with group information, by replacing the sampling of pairs of classes with the sampling of different group combinations. We compute $k$ for each model listed in § 3.

Furthermore, we evaluate the effect of model scale on amplification factor by varying the width of ResNet-18; evaluate various settings of weight decay; and evaluate the role of early stopping by computing amplification through training. Our choice to investigate these three parameters is motivated by their expected effect on simplicity bias. Following Kalimeris et al. (2019) we expect models to exhibit a stronger preference for simplicity earlier in training, which would often materialize when using early stopping. Weight decay is a common regularization technique intended to limit overfitting by penalizing excessively complex functions, and the role of width in over-reliance on spurious correlations is reported by Sagawa et al. (2020).

For a complementary test of the bias *against complexity*, we also try to push the model to choose a more complex solution. We enforce a Lipschitz constraint by applying a penalty on the norm of the gradients, a technique commonly used to stabilize discriminator training in GANs (Gulrajani et al., 2017). We add the following penalty term to our loss function $L$,

$$L' = L + \lambda(||\nabla_{\mathbf{x}} f(\mathbf{x})||_2 - C)^2 , \tag{3}$$

where $\nabla_{\mathbf{x}} f(\mathbf{x})$ is the gradient of the network's outputs with respect to its inputs, the penalty coefficient $\lambda = 10$, and $C$ determines the Lipschitz constraint: a low $C$ pushes the model towards simpler functions, and high $C$ towards more complex.

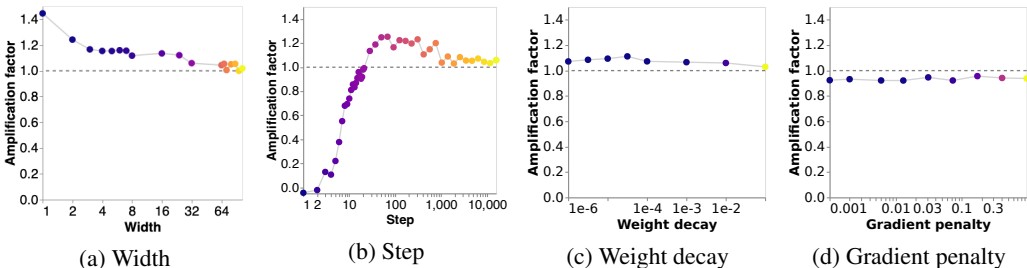

(a) Width      (b) Step      (c) Weight decay      (d) Gradient penalty

Figure 5: Design decisions influence amplification factor. All panels are a ResNet-18 on 30 sampled synthetic tasks based on CIFAR-100 (see fig. 3a). **(a)** Network width has a negative effect on amplification, reducing it to near 1. **(b)** $k$ peaks early in training before plateauing just above 1. **(c)** Weight decay has no effect on amplification. **(d)** Applying a gradient penalty to bias the model toward a $C$-Lipschitz functions works. See fig. S5 for $R^2$.

## 5.1 RESULTS

**Model architecture.** In fig. 4a, we illustrate the difference in amplification factor between two models, LeNet and ResNet-18. We find that the ResNet-18 amplifies disparity by a factor of $k = 1.19 \pm 0.12$. In contrast, LeNet diminishes disparity ($k = 0.84 \pm 0.04$), resulting in an observed disparity lower than expected. Thus, from this simple example we show that model choice influences difficulty amplification. Across the full suite of models (fig. 4b) we again see significant variation in amplification factors across the different models, with the certain models (e.g. LeNet) attenuating and the others (e.g. ResNet) amplifying. *However*, the simpler models all exhibit poor test accuracy averaged over the entire dataset (see fig. S3), offering a candidate explanation for the lack of amplification. These models may be too simple to learn the dataset at all, resulting in equally poor performance across all groups.

**Width.** However, within a specific architecture, increasing width seems to *reduce* amplification. Figure 5a shows the amplification factor for ResNet-18 rapidly decreasing to almost 1 (no amplification) as network width increases. These results align with those of Sagawa et al. (2020), who report that while overparameterization *typically* increases reliance on spurious correlations and harms worst-group error, this effect is *reversed* as groups become more balanced, such that increasing parameter count becomes helpful (Sagawa et al., 2020, e.g. fig. 6).

**Early stopping.** As training proceeds (fig. 5b), $k$ increases to a peak around 1.2 early in training, before decreasing to a plateau a little over 1. This highlights the important role of early stopping in amplifying disparity, particularly in light of prior work arguing that models learn more simple functions earlier in training (Kalimeris et al., 2019).

**Weight decay.** Figure 5c shows next to no effect of scaling the weight decay parameter. This is a surprising result, as our expectation is that applying stronger weight decay would further bias the model towards the simpler group, increasing amplification. One possible explanation is the sensitivity of the $\ell_2$ penalty to choices of model and dataset, as reported by Sagawa et al. (2019). While the purpose of our work is to introduce the notion of difficulty disparity and difficulty amplification, further research is needed to confirm the role of weight decay across various settings, and its interaction with other implicit regularization schemes.

**Gradient penalty.** In contrast, applying a penalty to the norm of the gradients, rather than the parameters, is sufficient to lower $k$ to below 1 for all values of $C$ considered here. This suggests that applying a gradient penalty to balance out the implicit bias towards simplicity may be a helpful strategy in combating difficulty disparity.

**Summary.** *High-performing models—those optimized for average test accuracy—consistently display difficulty amplification. This phenomenon is exacerbated by early stopping, but may be reduced using a gradient penalty.*

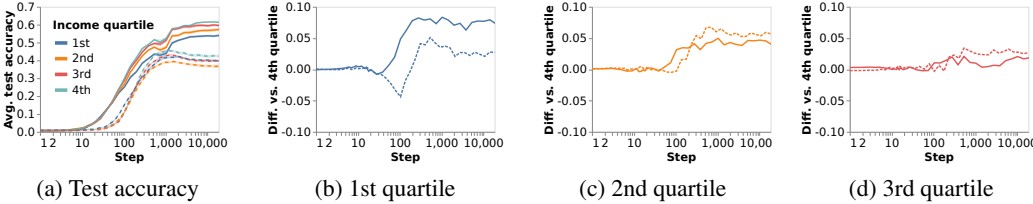

(a) Test accuracy  (b) 1st quartile  (c) 2nd quartile  (d) 3rd quartile

Figure 6: **(a)** Mean test accuracy for each income quartile on balanced Dollar Street. Dashed lines are single-group training; solid trained on entire dataset. **(b–d)** Performance disparity between each income quartile compared against the top income quartile (also the best performing). Dashed lines indicate estimated disparity $\tilde{d}$; solid indicate observed disparity $d$. This particular model amplifies the disparity between the highest and lowest income quartiles **(b)**, but reduces disparity between middle and top quartiles **(c–d)**. Error bars omitted for clarity.

## 6 DIFFICULTY AMPLIFICATION HAS REAL-WORLD IMPACT

We now evaluate difficulty disparity and amplification in a real-world case study. Dollar Street[1] is a dataset of geographically-diverse images spanning a broad range of household incomes. We use the labels associated with each image in a 138-class object classification task, where group information is household income quartile. We explicitly rebalance the dataset via subsampling to ensure each group is has the same number of data points. We evaluate models trained on each income quartile independently, and models trained on all quartiles together. We train single-layer FC networks on representations extracted from ResNet-18 pretrained on ImageNet-1K. See appendix D for details.

Figure 6a shows the test accuracies of the models trained on the whole dataset (solid lines), and the models trained on each income quartile separately (dashed). In fig. 6b–d, we compare the performance of the quartile upon which the models perform best, the highest income quartile, against the other three quartiles. In fig. 6b, we see that the gap between the highest and lowest income quartiles increases through training. Comparing estimated disparity ($\tilde{d}$; dashed line) with observed ($d$; solid) reveals evidence of difficulty amplification between the 1st and 4th quartile. In fig. 6c–d, we again observe that performance disparity between the middle and highest quartiles increases through training, though here we see slight disparity attenuation.

Thus, we present consistent evidence of performance disparity, such that the models continuously perform best on the highest income quartile. However, it is a mixed picture regarding difficulty amplification, with amplified disparity between the lowest and highest income quartiles, but slightly reduced elsewhere. There are a number of key takeaways. First, even using a dataset with a focus on diverse data collection, we still observe a bias of several percentage points towards the highest-income group. Second, explicitly rebalancing the dataset to ensure equal group sizes is not sufficient to remove this bias, nor can it be attributed to spurious correlations due to our cross-validation methodology. Third, difficulty amplification is present here, though not consistently throughout the dataset. In this case, by choosing to deploy this specific model we would magnify biases against low income households, less so against middle income households, all the while giving preferential treatment to high income.

**Summary.** *Example of difficulty disparity in social context: this model amplifies income disparity.*

## 7 DISCUSSION

**Auditing for bias.** At its heart, our work presents yet another way in which models exhibit bias and performance disparities across demographics. A frequent refrain in the ML community is that such disparities are the fault of the data, rather than algorithmic bias (Hooker, 2021). Indeed, a series of thorough audits have revealed that popular datasets under-represent minoritized groups (Shankar et al., 2017; Stock & Cisse, 2018; Buolamwini & Gebru, 2018; de Vries et al., 2019; Dulhanty & Wong, 2019; Wilson et al., 2019); reify harmful associations and perpetuate stereotypes

---
[1] https://www.gapminder.org/dollar-street

(Bolukbasi et al., 2016; van Miltenburg, 2016; Garg et al., 2018; Dixon et al., 2018; Birhane & Prabhu, 2021; Raji & Fried, 2021); and operationalize concepts such as gender and race in a way that applies a veneer of "objectivity" to socially-constructed and culturally specific concepts (Keyes, 2018; Paullada et al., 2021; Denton et al., 2021; Raji et al., 2021). Fixing these issues at the level of the data may not even be possible, for example it is often undesirable to collect the demographic information needed to ensure balance in the first place (Veale & Binns, 2017; Andrus et al., 2021; Hooker, 2021). That being said, acknowledging issues with our use of data does not absolve all that comes after, as exemplified by bias amplification (Zhao et al., 2017; Wang & Russakovsky, 2021; Hall et al., 2022). Here, in support of the role of *post-training* audit, we choose the setting where the data is "perfect", in that it is both explicitly balanced, and groups and labels are decorrelated. The variability of both difficulty disparity and amplification from model to model is a strong reminder that both those who develop and deploy ML systems must take action to ensure their fairness.

**Fairness definitions.** By discussing issues of bias and disparity, we engage in a broader discussion about fairness in ML systems. Here, we follow others in focusing on the performance gap between groups (Dwork et al., 2012; Hardt et al., 2016; Woodworth et al., 2017; Agarwal et al., 2018; Khani et al., 2019; Goyal et al., 2022b), though an alternative approach would be to focus explicitly on worst-group performance instead (Mohri et al., 2019; Sagawa et al., 2019; Zhang et al., 2020). Others rely upon counterfactual fairness (Kusner et al., 2017; Kilbertus et al., 2017; Loftus et al., 2018), according to which a "fair" system reaches the same decision on two otherwise identical individuals belonging to different protected groups, though this draws increasing criticism due to its requirement that concepts such as race or gender both be well-defined (Benthall & Haynes, 2019) and can be changed while only *minimally* impacting other attributes (Hu & Kohler-Hausmann, 2020; Hanna et al., 2020). Our aim in this work is not to use a metric by which to deem systems fair or unfair, but to highlight the possible role of model bias—in this case, due to preference for simplicity—that will have subsequent fairness impacts. Even assuming a satisfactory yardstick by which to measure, and a model accordingly deemed fair, fairness is of course not necessarily implied. When situated within a broader societal context, any model can be put to harmful use, and it is a common pitfall of the ML community to narrowly situate our work inside neatly-defined abstractions (Selbst et al., 2019).

**Spurious correlations.** Similarly, a key ambition of our work is to push research into sources of bias outside of the typical characterizations: spurious correlations and under-representation. Indeed we suggest that reducing the study of model bias to these two dimensions is an instance of excessive abstraction through formalization (Selbst et al., 2019). By focusing on the settings where these issues are resolved, we hope that future research can take a more nuanced look at the biased behavior of models where not *obviously* the result of a data issue. A plausible outcome of this kind of research could be that in certain situations ML might not be appropriate *at all*, if we can't guarantee that the system won't develop unpredictable and hidden biases.

## 8 CONCLUSION

We have argued that what a model finds difficult is not simply a function of the data, but a function of both model and dataset. This is particularly a problem in a fairness context if difficulty is correlated with group information. We have found that certain models further amplify difficulty disparity, resulting in observed difficulty disparity over and above estimated difficulty disparity, as a result of the bias of certain models towards easy examples. Difficulty amplification varies with model architecture, model scale, training time and regularization strategy, and seemingly innocuous design decisions can have a substantial and counter-intuitive impact. Finally, we have shown how difficulty disparity and amplification take place in the Dollar Street setting, where our simple model is biased against images in the low income quartile. Taken together, our results highlight the key role of the model—above and beyond the dataset—in creating group disparities.

**Limitations.** Our primary aim is to further highlight the key role of the model in accuracy disparity. We do however assume access to group information for audit purposes, which may not be available in many realistic scenarios, nor desirable to obtain. We intentionally choose to explore the balanced dataset setting, though separating difficulty disparity from other sources of bias may be difficult in practice. Future work may seek to explore a broader array of model families, and a more detailed investigation of the role of different regularization techniques.

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

## A    PLS METHODOLOGY

Given $m$ classes, we extract $d = m\frac{m-1}{2}$ binary accuracies corresponding to each possible pairing, and convert them to rank orders. We repeat this for each of the $n$ models under consideration, resulting in a $d \times n$ matrix of difficulty ranks. We construct a $d \times 1$ data difficulty matrix from the cosine distance between the mean of each class.

We fit a partial least squares regression model to both the model difficulties and the data difficulties using *scikit-learn* (Pedregosa et al., 2011). For visualization purposes we use a single component, though in subsequent tests we find no difference in model fit when increasing the number of components. We evaluate how well the data difficulty is explained by the model difficulty using $R^2$.

## B    MEASURING DIFFICULTY

In this work we *choose* to measure difficulty using cross-validated test accuracy, averaged over all samples in a group or class. Recent works have investigated alternative methods for quantifying model-specific example difficulty, including loss (Arazo et al., 2019; Han et al., 2018) and prediction disagreement between models (Simsek et al., 2022), mini-batches (Chang et al., 2017), and throughout training (Toneva et al., 2019; Swayamdipta et al., 2020). Hooker (2021) identifies samples that are often forgotten after compression. Applying these sample-level measures to evaluating group-level difficulty disparity remains an interesting future direction.

## C    MODEL ARCHITECTURE AND HYPERPARAMETERS

### C.1    SVM

For the SVM we use an RBF kernel with hyperparameters $C = 1.0$ and $\gamma = \frac{1}{3072}$, using the *scikit-learn* implementation.

### C.2    NEURAL NETWORKS

All models are implemented in *PyTorch* (Paszke et al., 2019) with *TorchVision*. Models are trained to minimize cross-entropy loss using SGD with learning rate 0.01, momentum 0.9, weight decay of 0.0001 for 500 epochs with batch size 128.

**FC.** The fully-connected networks are either 3 or 5 hidden layers with 256 units and ReLU activation. Batch normalization is applied to the inputs.

**LeNet.** LeNet (Lecun et al., 1998) is a simple CNN, with two convolutional layers interleaved with max pooling, three fully-connected layers, and ReLU activation function.

**AlexNet.** AlexNet (Krizhevsky et al., 2012) is a deeper CNN, with five convolutional layers interleaved with max pooling, three fully-connected layers, and ReLU activation function.

**ResNet-18.** We use the variable-width ResNet (He et al., 2016) implementation of (Sagawa et al., 2020).

**SSL.** For both SSL models, we extract final-layer representations for each data point from an SSL-pretrained model. We pass these representations through a 1-layer FC network as described above. Representations are extracted from one two models. The first is from a ResNet-50, pretrained with SimCLR (Chen et al., 2020) on ImageNet-1K (Russakovsky et al., 2015). The second is from a RegNet-128Gf model (Radosavovic et al., 2020) trained with SwAV (Caron et al., 2020) on 1 billion public images from Instagram (Goyal et al., 2022a). Representations were extracted using VISSL (Goyal et al., 2021) from models publicly available in the model zoo.

# D    DOLLAR STREET EXPERIMENT

## D.1    DATASET

Dollar Street[2] is a dataset of geographically-diverse images spanning a broad range of household incomes. Dollar Street comprises 23724 RGB $480 \times 480$ images of objects and people in everyday environments around the world, each associated with one of 138 class labels. For our purposes, we discard geographic information and use income quartiles as group label.

Throughout this work, we have endeavored to remove bias resulting from group imbalance and label/group correlation. However, in the Dollar Street example we introduce an additional possible source of bias via ImageNet-1K pretraining. ImageNet-1K significantly under-represents many social groups (Dulhanty & Wong, 2019) and geographies (Shankar et al., 2017; de Vries et al., 2019), and exhibits harmful associations between race and certain class labels (Stock & Cisse, 2018). Geographic under-representation is a plausible reason for income-quartile difficulty disparity. This, however, cannot explain difficulty amplification.

## D.2    MODEL

We follow the representation extraction method outlined in appendix C.2, though we use the representations from supervised learning models rather than SSL. Specifically, we extract from a ResNet-50 trained with supervised learning on ImageNet-1K (Russakovsky et al., 2015). We train a single layer fully-connected network of varying width, following the standard SGD training regime specified above.

# E    CALCULATING AMPLIFICATION FACTOR WITH LINEAR REGRESSION

Given a vector of estimated accuracy disparities $\tilde{\mathbf{d}}$, and a vector of observed accuracy disparities $\mathbf{d}$, we estimate the amplification factor using OLS linear regression.

In our synthetic setup, we additionally control for the effect of confounds including within-class separability (e.g. *aquatic mammal / medium mammal* in fig. 3a), and diagonal separability (e.g. *medium mammal / flower*), by including them as nuissance regressors. Our full model is of the form:

$$\mathbf{d} = X\beta + \epsilon, \quad X = \begin{pmatrix} \tilde{\mathbf{d}} \\ \mathbf{s}_{a,0}^{a,1} \\ \mathbf{s}_{b,0}^{b,1} \\ \mathbf{s}_{a,0}^{b,1} \\ \mathbf{s}_{a,1}^{b,0} \end{pmatrix}, \tag{4}$$

where amplification factor is the first parameter, $k = \beta_0$, and $\mathbf{s}_{a,0}^{b,1}$ is the separability (i.e. accuracy) between group $a$, label 0 and group $b$, label 1. We use Python *statsmodels* (Seabold & Perktold, 2010) to fit the model.

---

[2]https://www.gapminder.org/dollar-street

## F SUPPLEMENTARY FIGURES

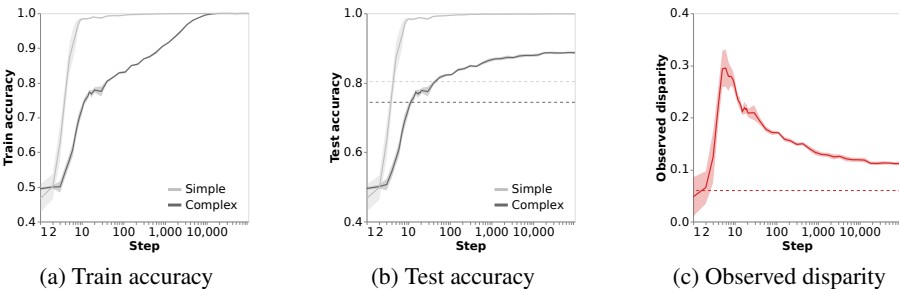

(a) Train accuracy  (b) Test accuracy  (c) Observed disparity

Figure S1: Accuracy and disparity for ResNet-18 on Fashion MNIST (Xiao et al., 2017). Experiment design is identical to fig. 3. Simple group is *Trouser/Sneaker*; complex group is *T-Shirt/Shirt*. Results align with those observed on CIFAR-100. **(a)** Train accuracy for complex group is learned slower but both groups reach perfect train accuracy. **(b)** However, test accuracy for complex group is persistently lower. Dashed lines are binary accuracy from single-group training. **(c)** Observed disparity $d$ peaks early in training but large gap persists after training. Red dashed line is estimated disparity $\tilde{d}$: observed disparity above this line indicates *amplification*. Shaded area is standard error over 10 runs each with different train/test split.

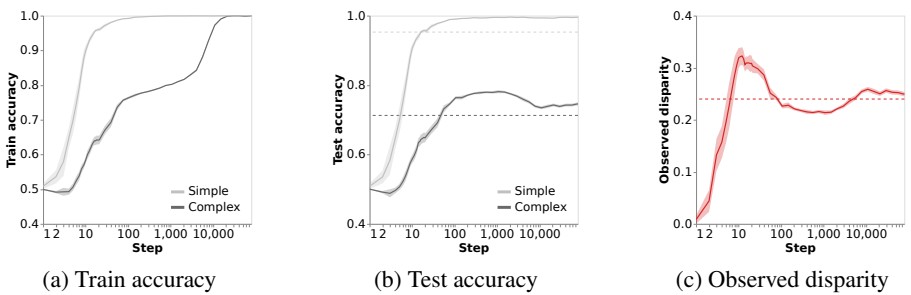

(a) Train accuracy  (b) Test accuracy  (c) Observed disparity

Figure S2: Accuracy and disparity for ResNet-18 on EMNIST letters (Cohen et al., 2017). Experiment design is identical to fig. 3. Simple group is *Q/X*; complex group is *I/L*. Results align with those observed on CIFAR-100. **(a)** Train accuracy for complex group is learned slower but both groups reach perfect train accuracy. **(b)** However, test accuracy for complex group is persistently lower. Dashed lines are binary accuracy from single-group training. **(c)** Observed disparity $d$ peaks early, drops below estimated disparity during training, and then stabilizes at slight amplification.

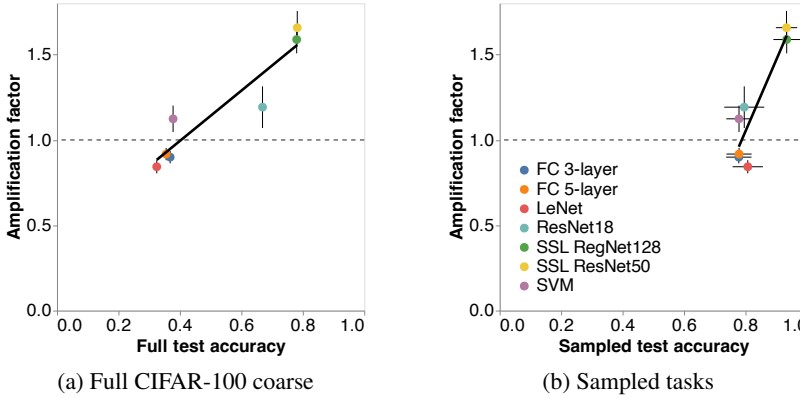

(a) Full CIFAR-100 coarse

(b) Sampled tasks

Figure S3: Amplification factor $k$ as function of average test accuracy on **(a)** the full CIFAR-100 coarse dataset, and **(b)** the 30 sampled tasks used for computing amplification factor. **Choosing a higher accuracy model, e.g./ an SSL model, would increase amplification.** Vertical bars are standard error of the coefficient $k$. Horizontal bars (barely visible in left panel) are standard deviation of test accuracy over (a) 10 seeds and (b) 10 seeds and 30 tasks. Solid black line fit with linear regression. Dashed gray line is $k = 1$.

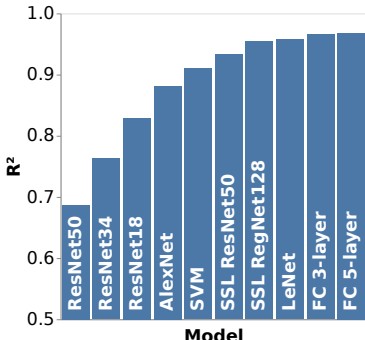

Figure S4: $R^2$ values for linear regression calculation of amplification factor, for various models, corresponding to fig. 4

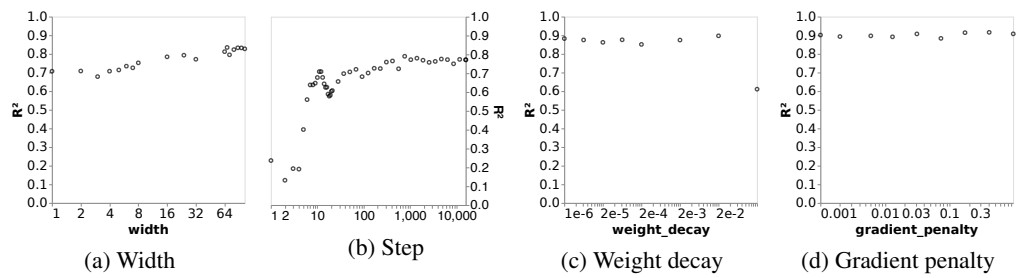

(a) Width

(b) Step

(c) Weight decay

(d) Gradient penalty

Figure S5: $R^2$ values for linear regression calculation of amplification factor, corresponding to fig. 5.

