# OpenReview forum: "Simplicity bias leads to amplified performance disparities"
_ICLR.cc/2023/Conference — Submitted to ICLR 2023_

### Official Review · Reviewer_Reqo · 2022-10-20

**Confidence:** 2
**Correctness:** 3
**Technical Novelty And Significance:** 2
**Empirical Novelty And Significance:** 2
**Recommendation:** 5

**Clarity, Quality, Novelty And Reproducibility:**

While most of the writing is clear, I found the following parts a bit confusing:

1. The paper could dedicate a separate subsection/paragraph to specify data-only vs data+model measures of difficulty. The 'PLS analysis' section could be re-titled to indicate what it is (data-only vs data+model) instead of mentioning the technique.

2. It is unclear how the class means are computed. If they are computed in the embedding space, isn’t that model-dependent?

3. I did not find the terms ‘observed’ and ‘estimated’ intuitive and had to keep looking back at the definitions. Specifying the intuition behind the terms or changing them to more intuitive terms could improve readability.

As far as I am aware, the measure of difficulty amplification is novel and the experiments seem reproducible.

**Strength And Weaknesses:**

**Strengths**

[S1] The paper provides a general, intuitive measure for difficulty amplification and the experiments show that this is dependent on the models and not just data.

[S2] If group labels are provided, then the paper can be used by newer works to measure difficulty disparities.

**Weaknesses**

[W1] This work requires group labels, which may not always be available. Recent works on debiasing techniques have realized that this is an unrealistic assumption, so use implicit measures of difficulty e.g., Learning From Failure, Just Train Twice etc. It is unclear how this work could leverage the implicit measures to compute amplification.

[W2] Group underrepresentation is a real-world problem. The model unfortunately does not consider this setting. Furthermore, ‘underrepresentations’ may not be the only source of spurious correlations e.g., even with balanced groups certain factors in the group (not necessarily signal) may be easier to exploit. So, the claim of ‘absent correlations between group labels and target labels’ may require further verification.

[W3] Weight decay is generally used to regularize the complexity of the model (to prevent overfitting), yet the paper surprisingly finds ‘next to no effect of weight decay’. Could you elaborate on this finding? Would it hold in general? One interesting experiment would be to analyze the models from [1], where the authors showed that with increased weight decay (decreasing overparameterization) even simple re-weighting could help debias models.

[1] Sagawa, Shiori, et al. "An investigation of why overparameterization exacerbates spurious correlations." International Conference on Machine Learning. PMLR, 2020.



**Summary Of The Paper:**

The paper introduces the notions of difficulty disparity and difficulty amplification, which is the ratio of change in the accuracy gap between different groups (easy vs hard groups) when trained in a combined manner versus when trained on those groups individually. The experiments show different models have different difficulty amplification, and it is not just a data-only issue.

**Summary Of The Review:**

While the measure might be useful, I think the paper needs to address at least these shortcomings: a) study on settings without group labels since that is more realistic b) confirm the weight decay finding better (perhaps using the underrepresented groups) to be accepted.

---

> ### Author Response · Authors · 2022-11-17
> **Response to reviewer Reqo**
>
> We really appreciate all your considered and helpful comments. Here are a few responses to your points.
>
> ### On availability of group labels:
>
> As we note in [our response to reviewer 9g7M](https://openreview.net/forum?id=mAWJpM7S21-&noteId=yXtf8-ERDA), we ourselves raise (in section 7) that demographic attributes are not always available, and in many real-world scenarios it is undesirable to even collect sensitive personal information.
>
> Broadly, however, we would just like to emphasize that our aim here is not to introduce a metric for use in model performance audits, or to present a bias mitigation solution (e.g. Just Train Twice, etc) but instead to introduce the concepts of difficulty disparity and amplification, and to highlight the fact that they are subtle, nuanced, and challenging to identify. We see the existence of difficulty disparity and amplification as proof of the necessity of post-training model audit, and to call for further research into how models can introduce and amplify performance disparities. Our work relies mainly on empirical experiments, though its contributions are not intended to be solely practical in nature.
>
> ### On group imbalance:
>
> We also discuss concerns about real-world group imbalance in [our response to reviewer 9g7M](https://openreview.net/forum?id=mAWJpM7S21-&noteId=FJI9vAs62-h).
>
> We totally agree that group imbalance is a real-world problem, though it’s worth breaking  this out into two possible scenarios. First, there is the scenario where there is genuine imbalance in the population distribution, for example global household income, or global population distribution by continent. Second, there is the scenario where the imbalance is the result of biased sampling practices, such as ImageNet’s over-representation of high-income households from North America and Europe. Our work is not designed to approach the first scenario, but the second. It is our ambition to demonstrate that, even if biased sampling practices were resolved, bias and performance disparities are still possible. Our aim is to present a critical response to the common suggestion that improving sampling practices, or collecting more data, is sufficient to create fair systems.
>
> ### On weight decay:
>
> Based on your feedback, we’ve extended section 5 to comment further on the role of weight decay.
>
> Indeed, we also found the limited effect of weight decay surprising: our expectation was that increasing weight decay would further bias the model in favor of the simpler group, further exacerbating difficulty amplification. That said, we do find some evidence of the opposite effect: that applying the gradient penalty to enforce a higher Lipschitz constant does seem to reduce difficulty amplification.
>
> We think both of these ideas are particularly interesting when interpreted in the context of Sagawa et al’s work [1]. While Sagawa et al. find that weight decay increases the reliance on spurious correlations (though notably it appears to vary with dataset and model), we find no effect on difficulty amplification. We read this as further evidence that the effect we describe is not the result of  subtle spurious correlations at the pixel- or feature-level.
>
> ### On data-only versus data+model difficulty measures:
>
> Thanks for the clarifying suggestion on how to improve section 3. We’ve updated the section accordingly.
>
> To answer your question about class means: we use the class means of the raw data, not of the embeddings. We’ve updated appendix A to make this clearer.
>
> [1] Sagawa, S., Koh, P. W., Hashimoto, T. B., & Liang, P. (2019). Distributionally robust neural networks for group shifts: On the importance of regularization for worst-case generalization. arXiv:1911.08731

---

> > ### Comment · Reviewer_Reqo · 2022-11-28
> > **Post Rebuttal Response**
> >
> > Thank you authors for responding to my concerns.
> >
> > The edits address some of my concerns, so I am increasing the score to a 6.
> >
> > However, I think a thorough study of weight decay, perhaps using population imbalance is necessary for a clear acceptance.

---

> > > ### Author Response · Authors · 2022-12-01
> > > **Thanks for your response**
> > >
> > > Thanks for the response, and we are glad to see that your concerns have been addressed.
> > >
> > > We also agree an additional study of weight decay, and its relationship to data balance, would be a very interesting extension, particularly in light of Sagawa et al.’s work as we discuss in the comment above.
> > >
> > > Our hypothesis is that the effect of weight decay would increase as correlation between group label and target label increases. If this hypothesis is correct, it would simultaneously account for both our finding (i.e., no effect of weight decay **when there are no spurious correlations**), and Sagawa et al.’s finding (i.e., weight decay increases overreliance on spurious correlations, **if spurious correlations are present**). We’re currently in the process of designing additional experiments to test this hypothesis.

---

### Official Review · Reviewer_9g7M · 2022-10-21

**Confidence:** 4
**Correctness:** 2
**Technical Novelty And Significance:** 2
**Empirical Novelty And Significance:** 3
**Recommendation:** 5

**Clarity, Quality, Novelty And Reproducibility:**

Overall, I believe the main message of the paper is clearly conveyed in the manuscript, albeit limited, the contribution is novel to the best of my knowledge, and most of the information required for reproducing the experiments is reported in the Appendix. However, as mentioned in the previous section of my review, I have several concerns regarding this submission, which I detail below along with questions and suggestions.

- A major flaw I see in this work is the fact that the authors do not provide any evidence to support the claim that the experimental setting in the CIFAR-100 experiments is indeed absent of correlations between group labels and target labels (as claimed in the manuscript in the beginning of Section 4.2). Moreover, it is also important to emphasize that different model classes can potentially capture different spurious correlations which makes it even more difficult to make such a claim. This point exemplifies reasons why I rated the “Correctness” of this submission as 2.

- The experiments suggest that SGD-trained neural networks are biased towards less complex solutions. Although I found this conclusion interesting, I’m concerned with how generalizable this finding is to other optimizers often used in the literature and in practice. More specifically, would the conclusions hold in case we train a neural network with Adam or RMSprop? If not, then switching SGD by another optimizer would already be enough to avoid the problem of amplifying disparity? If so, is this problem even worth studying? I believe that all the aforementioned questions render the motivation for this work unclear. This point exemplifies reasons why I rated the “Technical Novelty And Significance” and “Empirical Novelty And Significance” of this submission as 2.
-
I found the introduced notion of difficulty amplification based on accuracy of models trained on specific groups very limited given that in real-world scenarios it is likely that there will be groups for which data will be scarce (e.g. a rare disease). In case one wants to reproduce such an analysis to diagnose the amplification disparity of a given model class, this makes the possible application scenarios very limited. This point also exemplifies reasons why I rated the “Technical Novelty And Significance” and “Empirical Novelty And Significance” of this submission as 2.


- The findings reported in Section 5 are underexplored and lack depth in the analysis of the results. The authors only report which of the considered aspects affected or not the amplification of difficulty disparity without providing any hints on possible reasons that would explain those findings. I also found it surprising that weight decay does not have an effect on difficulty amplification since it is tightly related to model complexity. This point also exemplifies reasons why I rated the “Technical Novelty And Significance” and “Empirical Novelty And Significance” of this submission as 2.


Minor:
- Typo in page 4: difficuly -> difficulty


**Strength And Weaknesses:**

Strengths:
- The paper adds evidence to show that biased predictions are an effect of both datasets and models biases, a often discussed question within the machine learning community;
- The paper is clearly written;
- The authors made a clear effort to convey their message by providing pictures that clearly illustrate the issues tackled in the paper and summarizing the main findings of each experiment.

Weaknesses:
- The paper contributions are mostly empirical and the scope of the experiments is quite narrow, which makes it difficult to conclude how robust the findings are to other common settings such as neural networks trained with optimizers different from SGD;
- Critical claims to support the findings are not well explained and supported. Importantly, the authors claim that they perform experiments in a setting where models do not pick up on spurious correlations between input and labels, but in no way this claim is supported in the manuscript.
- The proposed measure of difficult amplification relies on training predictors for each particular group in the dataset. I believe this is not practical in many real-world settings where examples from particular groups are very scarce, making it difficult to train and evaluate specialized models.
- Computing the proposed measure of difficult amplification further requires knowledge about group labels so that the dataset can split. This information is not always available in practice and / or shouldn’t even be taken into account when collecting a dataset.


**Summary Of The Paper:**

This work proposes a notion of difficulty disparity between subsets of a dataset and claims that neural networks parameterize functions that better separate groups with lower difficulty. The claim is then validated empirically via a set of experiments on CIFAR-100 and the Dollar Street datasets. Results indicated that difficulty amplification is affected by factors such as model architecture and early stopping.

**Summary Of The Review:**

This submission is clearly written and aims at shedding light on interesting and relevant aspects of SGD-trained models. My major concerns are related to the lack of support for central claims of the narrative in the paper, the generalization of the findings, and lack of depth in the experimental analysis, which was critical to my score given that the contributions of this work are mostly empirical. All in all, I found that the flaws in the current version of this submission outweigh its merits by a large margin and I believe this manuscript is not ready for publication yet.

---

> ### Author Response · Authors · 2022-11-17
> **Response to reviewer 9g7m (part 1)**
>
> Thanks for your effort in putting together such a thorough and helpful review.
>
> ### On empirical experiments:
>
> Based on your feedback, we’ve extended the analysis of model architecture to include additional architectures (see fig. 4b) and repeated our experiments on a selection of new datasets including EMNIST letters and Fashion MNIST, to demonstrate the generality of our findings (see appendix F).
>
> We agree—of course—that our contribution is mostly empirical - that was the point! We have gone to great lengths to design careful experiments that show the connection between simplicity bias (as reported elsewhere, e.g. [1,2,3]) and performance disparities between groups. While our work is empirical, our experiment design is entirely novel (see sections 4 and 5), as is the phenomenon under investigation.
>
> Throughout the course of the paper, we evaluate a range of different models, two different (opposing) regularization strategies in the form of weight decay and the gradient penalty, the effect of early stopping, and the effect of network scale.
>
> ### On SGD and its variants:
>
> When we say “SGD-trained”, we are referring broadly to SGD and other variants of gradient-based optimizers, including RMSprop and Adam as you mention. At present, we have no reason to believe these other optimizers would behave any differently in this setting, particularly in light of emerging evidence that simplicity bias persists across other algorithms [4,5,6].
>
> ### On controlling for spurious correlations:
>
> We control for spurious correlations in two ways. First, we use stratified subsampling to explicitly create datasets where there is an equal number of datapoints in each group to remove any obvious skew towards one particular group. Second, every result is the average over 10 runs, where each run is a different model initialization, different train/test split, and different stratified subsample. By averaging over subsamples, we control for the effect of covariate shift, which is typically what is referred to as spurious correlations (e.g. where the background is correlated with labels in the train set, but not in the test set).
>
> The reviewer is of course correct, it remains a possibility that there are pixel-level or feature-level spurious correlations between certain data points and certain labels in the dataset as a whole. Likewise in any experiment, it’s possible that there are unknown confounding factors. However, we believe the above approach to be sufficient to control for the most significant confounding factors, and those that are most likely in reality: group <> label correlation. Perfectly controlling for micro-level correlations is challenging without constructing fully synthetic data points: something we’d love to see but is outside the scope of this current work.
>
> Finally, the fact that these micro-level (e.g. pixel, feature) spurious correlations are subtle and hard to identify is a strong argument in favor of post-training model-specific performance audits, which is really the ultimate point of our paper.
>
> ### On group imbalance:
>
> A key ambition of our paper is to push the consideration of group disparity into a more nuanced setting, rather than focusing on the overly-simplified imbalanced dataset case. It is a common retort to blame problems of bias on obviously biased data, e.g. where datasets have significantly imbalanced group proportions, and that if we could somehow “fix” the data these issues would be mostly resolved. By focusing on the scenario where these issues are resolved, we show that bias and performance disparities still emerge, and create real world impacts.
>
> Moreover, while there are some real-world settings where group scarcity is the reality (e.g. healthcare and rare diseases), there are other scenarios where imbalance arises simply from biased sampling (e.g. ImageNet’s known bias towards high-income North American and European households). “Fixing” the dataset is *sometimes* possible, but our paper shows it’s still nowhere near enough to confidently remove disparity.
>
> [1] Valle-Perez, G., Camargo, C. Q., & Louis, A. A. (2018). Deep learning generalizes because the parameter-function map is biased towards simple functions. arXiv:1805.08522.
>
> [2] Kalimeris, D., Kaplun, G., Nakkiran, P., Edelman, B., Yang, T., Barak, B., & Zhang, H. (2019). Sgd on neural networks learns functions of increasing complexity. NeurIPS.
>
> [3] Mingard, C., Skalse, J., Valle-Pérez, G., Martínez-Rubio, D., Mikulik, V., & Louis, A. A. (2019). Neural networks are a priori biased towards boolean functions with low entropy. arXiv:1909.11522.
>
> [4] Shah, H., Tamuly, K., Raghunathan, A., Jain, P., & Netrapalli, P. (2020). The pitfalls of simplicity bias in neural networks. NeurIPS.
>
> [5] Qian, Q., & Qian, X. (2019). The implicit bias of adagrad on separable data. NeurIPS.
>
> [6] Wang, B., Meng, Q., Chen, W., & Liu, T. Y. (2021, July). The implicit bias for adaptive optimization algorithms on homogeneous neural networks. ICML.

---

> > ### Author Response · Authors · 2022-11-17
> > **Response to reviewer 9g7m (part 2)**
> >
> > ### On the availability of sensitive attributes / group information:
> >
> > We are in complete agreement that demographic information is not always available, and often may be undesirable to collect. We raise this point ourselves in section 7, under “Auditing for bias.”
> >
> > By introducing difficulty disparity and amplification, our intention is not to add a new evaluation metric that should be adopted during audit (which would, yes, necessitate the use of sensitive information). Instead, a key point of our paper is to show that difficulty disparity and amplification exist, and that they are *not visible* at the point of dataset audit, and that simply balancing the data is not sufficient to remove it. The existence of difficulty disparity and amplification calls for more research, and a more nuanced, less artificial framework in which to think about bias.
> >
> > Finally, for the purposes of future empirical investigations such as ours (i.e. not practical audits), and future work that seeks to further investigate the causes and mitigation of difficulty disparity, a number of datasets for fairness research that are annotated with sensitive information are now available, including e.g. FairFace, OpenImages MIAP, Casual Conversations and Dollar Street.
> >
> > ### On the variability of amplification factor:
> >
> > Thanks for your helpful critique of section 5, and our exploration of the effect of various design decisions, including the effect of network width, training time, weight decay and gradient penalty. These experiments were selected because we expect each to play a part in further biasing the model either towards or away from simplicity. Based on your feedback,  we have now reworked and expanded section 5, in order to: a) better explain the motivations for testing the influence of these parameters, b) add further detail surrounding why we believe these results are important, and c) add a particular focus on the (lack of) effect of weight decay. We hope this helps to assuage your concerns about technical novelty, and would of course welcome further feedback.

---

### Official Review · Reviewer_eHoV · 2022-10-25

**Confidence:** 5
**Clarity, Quality, Novelty And Reproducibility:** Please see strengths and weaknesses.
**Correctness:** 3
**Technical Novelty And Significance:** 3
**Empirical Novelty And Significance:** 3
**Recommendation:** 5

**Strength And Weaknesses:**

Strengths:
- CIFAR-100 setup. The synthetic variant of CIFAR-100 incorporates an intuitive notion of model-specific difficulty and clearly showcases that difficulty amplification can show up in standard image classification datasets.

- Amplification factor analysis. The analysis on how the amplification factor changes depending on standard design choices such as architecure, early stopping etc is insightful.

- The paper (especially the first few sections) is well written.


Weaknesses:
- Main finding not novel: The main finding of this paper (performance disparity is amplified when easy and hard groups are clumped in the dataset) is not new. This paper (https://proceedings.neurips.cc/paper/2020/hash/6cfe0e6127fa25df2a0ef2ae1067d915-Abstract.html) shows that simplicity bias hurts generalization when the dataset comprises a majority subpopulation of "easy" data points and a minority subpopulation of "hard" data points---the drop in accuracy stems from misclassifying the minority subpopulation, even though a model trained only on the minority subpopulations performs well. The authors should clearly contextualize the experiments in this paper vis-a-vis previous findings on this phenomenon.

- Section 3: The findings in this section contradict previous work. There are several works (e.g. http://proceedings.mlr.press/v119/hacohen20a/hacohen20a.pdf and references therein) that show that models with different architectures etc learn similar classifiers + classify points correctly over time in a similar order. On the other hand, this paper says that dataset difficulty is different for different models. So, it would be nice to reconcile these findings with previous works. Is this because the models compared in this paper are too different (e.g, linear models and CNNs)? Also, PLS analysis hard to understand: The paper needs to introduce PLS briefly before it introduces the findings. The way it is written right now is hard to parse.

- Analysis on disparity over time. It is not clear why the paper focuses on difficulty amplification over time. What is the practical implication of understanding how difficulty amplification varies over time? Having some clear motivation would be useful, right now it feels a bit ad-hoc.

- Section 6. This section is too terse. I think the experiment setup and the analysis should be more in-depth given that this is the only experiment showcasing this phenomenon in practice. Most of the discussion can be deferred to an appendix to make room for this.

Overall, this paper studies an important problem that is not well understood. However, my major concerns are (a) novelty---previous work on simplicity bias has analyzed this phenomenon, (b) section 3 claims contradict previous findings on this topic, (c) limited + unclear analysis on difficulty amplification in practice.

**Summary Of The Paper:**

This paper focuses on the problem of difficulty amplification: trained models exhibit consistent differences in performance on groups that are "easy" and "difficult" even when there are no obvious spurious correlations or dataset imbalance. In particular, they show that the  simplicity bias is a major reason for difficulty amplification. First, they show that data difficulty is model-specific (different models find different aspects of the dataset difficult). Then, the paper shows that models trained on a variant of CIFAR-100 (with easy and hard groups) exhibit difficulty amplification. Section 5 focuses on understanding how the amplification varies as a function of design choices such as model architecture, model width, early stopping etc. Section 6 shows that difficulty amplification can be an issue in practice as well.

**Summary Of The Review:**

Please see strengths and weaknesses.

---

> ### Author Response · Authors · 2022-11-17
> **Response to reviewer eHoV (part 1)**
>
> Thanks so much for your feedback.
>
> ### On the novelty of our central finding:
>
> We really love Shah et al.’s work [1]. Indeed, it partially motivated our exploration of the pitfalls of simplicity bias in various settings, in particular those outside of the common imbalanced data + spurious correlations setup.
>
> We’d like to highlight three crucial differences between our work and Shah et al.’s. First, while the reviewer is correct that Shah et al. have previously shown that simplicity bias can be harmful, they do this in an imbalanced data setting. As such, they show that when there is a different between “hard” and “easy”, and when the minority population is hard, the model priorities the easy at the expense of the hard. In our work, we show that this is not a function of group imbalance, but a fundamental result of a preference for simple, even in the balanced setting.
>
> Second, Shah et al. show how simplicity bias leads to a reliance on simple *features*, rather than complex. In Shah et al.’s work, this refers to spurious correlations, such as a simple yet undesirable background correlation (see point (i) “Lack of robustness” in Shah et al.’s page 2). In our work, we show that simplicity bias leads to preferential treatment of a simpler group, in the absence of spurious correlations. When we say simpler group, this could be the result of decision boundary, features, or otherwise. Both our work and Shah et al. show how simplicity bias can be harmful, though we both investigate different ways in which this is the case.
>
> Third, while Shah et al. manually construct hard and easy groups, in our work difficulty is viewed as model-specific. This is a crucial distinction, because while it is easy to identify a minority group, it is hard to identify a model-perceived difficult group (particularly if we assume equal number of samples in each, as in our work).
>
> ### On PLS methodology:
>
> Thanks for the suggestion to clarify the PLS methodology in section 3. We provide full details of our methodology in Appendix A, but have now updated the section to better introduce PLS before reporting the results.
>
> ### On contradictions with previous work:
>
> Thank you for bringing Hacohen et al.’s [2] work to our attention. There are a few nuanced reasons why our work seems to disagree at first glance.
>
> First, while Hacohen et al. show that classifiers within the same architecture (trained with different initialisations and different batch samplings) exhibit high agreement, this is *not* the case with models of different architectures. The cross-architecture comparisons in Hacohen et al. (section 4) are actually few in number, but in general they show some agreement, *but not perfect*. For example, fig 8a shows that while AlexNet and ResNet agree on roughly 30k examples from the ImageNet validation set, but by the end of training ResNet and AlexNet disagree on 10k. Rightly, this is attributed to the fact that ResNet can continue to learn challenging examples while AlexNet struggles to improve past a certain point.
>
> We don’t dispute this, and indeed it aligns with the results we show in the fig 2b: architectures exhibit a certain amount of similarity, but *they also exhibit individual differences*. In Hacohen et al.’s work, this is the disagreement between ResNet and AlexNet. In our’s, it’s the non-perfect (i.e. <1) Tau in fig 2b. We’ve updated section 3 to clarify this point in relation to Hacohen et al.’s work.
> The important thing is that while there is some level of similarity, there is also nuanced difference, and that difference can and does matter because it translates into performance disparity.
>
> While drawing out broad similarities between architectures is tremendously interesting and valuable research, that doesn’t entail smoothing over performance heterogeneity. Roughly similar is not perfectly the same. In the context of fairness, this is a crucial point: any change in performance that aligns with demographic information is something to be carefully understood.
>
> [1] Shah, H., Tamuly, K., Raghunathan, A., Jain, P., & Netrapalli, P. (2020). The pitfalls of simplicity bias in neural networks. NeurIPS.
>
> [2] Hacohen, G., Choshen, L., & Weinshall, D. (2020, November). Let’s agree to agree: Neural networks share classification order on real datasets. ICML.

---

> > ### Author Response · Authors · 2022-11-17
> > **Response to reviewer eHoV (part 2)**
> >
> > ### On the effect of training time:
> >
> > Thanks for your suggestion. We have now updated section 5 to better motivate the choice to evaluate network width, training time, weight decay and gradient penalty.
> >
> > Broadly, we evaluate early stopping because it is a commonly-used regularization technique, and as such we would expect it to further bias the model towards the simpler group.
> >
> > The practical implication is, essentially, to be careful when using early stopping.
> >
> > ### On real-world applications:
> >
> > Thanks for your helpful constructive feedback about real-world impacts in section 6. We’ve now revamped section 6 to include a fuller analysis. We’d welcome your further feedback on this section.

---

> > > ### Comment · Reviewer_eHoV · 2022-11-19
> > > **Post-rebuttal response**
> > >
> > > Thank you, the rebuttal addresses my question on difference between this paper and Hacohen et al's work. Re: simplicity bias, I am not entirely convinced whether the novelty relative to Shah et al's work---group imbalance effect and model-specific difficulty---is significant enough at the moment, so I am keeping my score as is.

---

> > > > ### Author Response · Authors · 2022-11-29
> > > > **Re novelty of our contribution**
> > > >
> > > > Thanks for responding to our rebuttal. We just add to add a few points to clarify the novelty of our work with respect to Shah et al.
> > > >
> > > > 1. Shah et al. use the typical setting of imbalanced data. We show that simplicity bias is harmful **even without** imbalanced data.
> > > >
> > > > 2. Shah et al. focus on spuriously correlated features (e.g. background). We show that simplicity bias is harmful **even when** there are no obvious correlations between demographic information and label.
> > > >
> > > > 3. Finally, the definition of “difficulty” in Shah et al.’s work is hand constructed and general. In our work, we show that what counts as difficulty is specific to a given model, and varies from model to model.
> > > >
> > > > Concretely, our novel contribution is to show that simplicity bias leads to performance disparities in a fundamentally different scenario, and one that is underexplored by current work, including that of Shah et al.. In short, this is a new, and different, way in which simplicity bias can be harmful, that warrants future investigation. It is our aim to **introduce and characterise** this new form of model bias.
> > > >
> > > > By showing this, we demonstrate that conventional approaches that focus on spurious correlation and imbalanced data **are not sufficient** to explain and mitigate bias. This is a crucial difference and situates our work as related, yet significantly different, from prior work.

---

### Official Review · Reviewer_po3o · 2022-10-30

**Confidence:** 3
**Correctness:** 3
**Technical Novelty And Significance:** 2
**Empirical Novelty And Significance:** 2
**Recommendation:** 5

**Clarity, Quality, Novelty And Reproducibility:**

The paper is well written and easy to read. The novelty lies in the observation of “difficulty disparity”.

**Strength And Weaknesses:**

Strength

- The paper performs investigation of the task difficulty and reveals it is a function of both model and dataset.

Weakness
- The main argument of the paper (the task difficulty is not a function of the dataset itself but relies on the model as well) seems to be a known fact as different models have different capacity or generalization ability and their difficulty of solving a task also varies. For example, a deep CNN (e.g, ResNet-152) can perform better than a shallower one on ImageNet and thus the difficulty of solving ImageNet is different for different models.

- There is no explanation on how the model bias leads to different difficulty disparity. The model bias is also not clearly defined. The experiments are only based on a handful of different architectures/pre-trained models and it is not clear why these models are selected.

- The paper does not provide a solution for precisely estimating the difficulty disparity of a model. Given section 6, it is not still clear about the real-world impact of the difficulty amplification and what practical lessons can be learned from the observations.


**Summary Of The Paper:**

This paper explores how “difficulty” is model-specific, such that different models find different parts of a dataset challenging. They measure difficulty as a function of both model and data,  and they measure how difficulty affects performance disparity and quantify the effect of the simplicity bias. Their main observations are: 1. They identify difficulty disparity persists in the post-dataset audit setting. 2. They introduce a difficulty amplification factor to quantify how much a model exacerbates difficulty disparity. 3. They evaluate how choices including model architecture, training time, and parameter count impact difficulty amplification.


**Summary Of The Review:**

The paper presents an investigation on the association of task difficulty and model bias. However, the observations are not quite convincing.

---

> ### Author Response · Authors · 2022-11-17
> **Response to reviewer Po3o (part 1)**
>
> Thanks for your helpful and constructive feedback. We’d like to respond to a couple of points raised in your review.
>
> ### On the novelty of difficulty as function of model + data:
>
> First, the main argument of our paper is *not* that task difficulty is model-specific—we agree that that’s unsurprising, and of course some models are more appropriate for certain tasks. Instead, our main argument is that precisely *which parts* (classes, groups, datapoints) of the dataset are challenging is model-specific. This is a crucial difference: we aren’t concerned with task performance on aggregate over a dataset, but the heterogeneous performance disparities that exist between classes and groups.
>
> To continue with your deep vs. shallow CNN example, we agree that on average the deep CNN will likely exhibit better performance on ImageNet than the shallow CNN, *when averaged over the entire dataset*. However, what we can’t say with confidence is whether this performance gain will be uniformly distributed across all data points (unlikely), or whether it will vary; choosing a different model may have a positive impact on some classes, but that by no means precludes a negative impact on others. This is the central motivation behind difficulty disparity.
>
> ### On the connection between simplicity bias and difficulty disparity:
>
> Second, our intention in this work is to connect the extensive body of work on the inductive biases of SGD-trained models to a fairness context. It has previously been shown [e.g. 1,2,3] that neural networks exhibit a “simplicity bias” - i.e. they tend to learn simpler functions before more complex ones.
>
> It isn’t our aim to introduce simplicity bias, but to show how this can manifest in the form of disparate outcomes. Principally, we show this experimentally, through the careful design of experiments designed to causally manipulate the difference in (model-specific) complexity between groups (section 4, in particular 4.2 and 4.3). Our results show that when the model does perceive a difference in group difficulty, it will “double down” and further prioritize learning the simpler group (figs. 3c and 3d). We call this difficulty amplification, and we think it is simplicity bias in action.
>
> The evidence that this is simplicity bias is further confirmed by our experiments with early stopping and gradient penalty. Regularizing via early stopping is well known to bias models *towards* simpler solutions. Fig. 5b shows that difficulty amplification is at its peak early on in training, which we interpret as evidence of simplicity bias.
>
> Similarly, our experiments with the gradient penalty are also suggestive of simplicity bias. In this case, we use the gradient penalty to bias the model towards the more complex group, which equalizes performance, and reduces difficulty amplification (fig. 5d).
>
> We chose to evaluate on a range of different models to show that amplification—the extent to which a model *further prioritizes* what it finds simple—varies depending on model. This is further evidence in support of the post-training model audit. We selected a range of models to cover different architectures (e.g. fully-connected vs. convolutional), model depth, parameter count, and training regime (e.g. supervised vs. self-supervised). Based on your feedback, we have updated section 5 to better explain our choice of models, and included a few extra models in our comparisons.
>
> [1] Valle-Perez, G., Camargo, C. Q., & Louis, A. A. (2018). Deep learning generalizes because the parameter-function map is biased towards simple functions. arXiv:1805.08522.
>
> [2] Kalimeris, D., Kaplun, G., Nakkiran, P., Edelman, B., Yang, T., Barak, B., & Zhang, H. (2019). Sgd on neural networks learns functions of increasing complexity. NeurIPS.
>
> [3] Mingard, C., Skalse, J., Valle-Pérez, G., Martínez-Rubio, D., Mikulik, V., & Louis, A. A. (2019). Neural networks are a priori biased towards boolean functions with low entropy. arXiv:1909.11522.

---

> > ### Author Response · Authors · 2022-11-17
> > **Response to reviewer Po3o (part 2)**
> >
> > ### On estimating difficulty disparity, and practical lessons:
> >
> > We give a precise definition of estimated and observed disparity in section 4.1, and further detail on amplification factor in appendix E. Simply put, observed disparity involves auditing for differences in accuracy between groups.
> > Our aim by introducing difficulty disparity is not to introduce a new metric to track, such that all model audits should be calculating this. Instead, this paper introduces difficulty disparity to show how simplicity bias can lead to disparate outcomes, and highlights an important phenomenon that is so far underexplored.
> >
> > Note that we have chosen to evaluate disparity using accuracy, though in different settings alternative evaluation metrics may be more appropriate. A different evaluation metric can be easily swapped in place of accuracy in equations (1) and (2).
> > The are two key practical takeaways from our work:
> >
> > 1. Performance disparities emerge and persist during training, are model-specific, and are impossible to predict from a dataset audit. Thus, a post-training model audit is a must to ensure fair performance across groups.
> >
> > 2. One *reason* for these disparities is the simplicity bias, and our second takeaway, is that the negative consequences of simplicity bias deserve further empirical and theoretical investigation. In particular, we hope more work can investigate the role of model-specific difficulty, rather than focusing on (overly simplified) scenarios of imbalanced data or data with obvious spurious correlations between group and label. Moreover, as suggested by our investigation with the gradient penalty, our work adds to emerging evidence that regularization (i.e. biasing towards simple) is not always helpful.
> >
> > In the context of a fairness audit, such as during the development of a new model, measurements of persistent disparity (such as we introduce) can be applied to help mitigate potential model-induced biases. Nuanced understandings of the different types of bias (algorithmic, data, etc) being displayed are essential if we are to make progress in rectifying them.

---

### Author Response · Authors · 2022-11-17
**Thanks for the constructive feedback**

We would like to start with a huge thank you to all the reviewers for their detailed and useful reviews.

We’re pleased to hear that you find our paper well-written (9g7M, eHoV, po3o) and easy to read (po3o), and that you find the central ideas important (eHoV), interesting and relevant (9g7M) and useful (Reqo). We’re also grateful that you find the ideas we present intuitive (Reqo, eHoV) and insightful (eHoV).

We’ve responded in depth to each of your comments below, and have uploaded a new version of the manuscript with substantial revisions, which we hope will increase your confidence in our work.

Thanks again for your time and support.

---

### Decision · Program_Chairs · 2023-01-20

**Decision:**

Reject

**Justification For Why Not Higher Score:**

There was a consensus among reviewers that this paper should be rejected.

**Justification For Why Not Lower Score:**

N/A

**Metareview: Summary, Strengths And Weaknesses:**

This paper explores how different models may find different parts of the dataset difficult. There was a consensus among reviewers that this paper should be rejected. The main reasons for this were limited novelty (and underexplored connections) to prior work and limited experimental evidence. As a result, I recommend to reject this paper.